# Shake Table Tests on Scaled Masonry Building: Comparison of Performance of Various Micro-Electromechanical System Accelerometers (MEMS) for Structural Health Monitoring

**DOI:** 10.3390/s25041010

**Published:** 2025-02-08

**Authors:** Giuseppe Occhipinti, Francesco Lo Iacono, Giuseppina Tusa, Antonio Costanza, Gioacchino Fertitta, Luigi Lodato, Francesco Macaluso, Claudio Martino, Giuseppe Mugnos, Maria Oliva, Daniele Storni, Gianni Alessandroni, Giacomo Navarra, Domenico Patanè

**Affiliations:** 1Istituto Nazionale di Geofisica e Vulcanologia-Osservatorio Etneo, 95125 Catania, Italy; giuseppe.occhipinti@dica.unict.it (G.O.); giuseppina.tusa@ingv.it (G.T.); antonio.costanza@ingv.it (A.C.); gioacchino.fertitta@ingv.it (G.F.); luigi.lodato@ingv.it (L.L.); francesco.macaluso@ingv.it (F.M.); 2Department of Engineering and Architecture, University of Enna “Kore”, 94100 Enna, Italy; francesco.loiacono@unikore.it (F.L.I.); giuseppe.mugnos@unikorestudent.it (G.M.); maria.oliva@unikore.it (M.O.); giacomo.navarra@unikore.it (G.N.); 3Istituto Nazionale di Geofisica e Vulcanologia-Osservatorio Vesuviano, 80125 Napoli, Italy; claudio.martino@ingv.it; 4Wise Robotics, 00131 Roma, Italy; daniele.s@w1s3.com (D.S.); gianni@w1s3.com (G.A.)

**Keywords:** structural health monitoring, MEMS, shake table seismic tests

## Abstract

This study presents the results of an experimental investigation conducted on a 2:3 scale model of a two-story stone masonry building. We tested the model on the UniKORE L.E.D.A. lab shake table, simulating the Mw 6.3 earthquake ground motion that struck L’Aquila, Italy, on 6 April 2009, with progressively increasing peak acceleration levels. We installed a network of accelerometric sensors on the model to capture its structural behaviour under seismic excitation. Medium-to lower-cost MEMS accelerometers (classes A and B) were compared with traditional piezoelectric sensors commonly used in Structural Health Monitoring (SHM). The experiment assessed the structural performance and damage progression of masonry buildings subjected to realistic earthquake inputs. Additionally, the collected data provided valuable insights into the effectiveness of different sensor types and configurations in detecting key vibrational and failure patterns. All the sensors were able to accurately measure the dynamic response during seismic excitation. However, not all of them were suitable for Operational Modal Analysis (OMA) in noisy environments, where their self-noise represents a crucial factor. This suggests that the self-noise of MEMS accelerometers must be less than 1 µg/√Hz, or preferably below 0.5 µg/√Hz, to obtain good results from the OMA. Therefore, we recommend ultra-low-noise sensors for detecting differences in the structural behaviour before and after seismic events. Our findings provide valuable insights into the seismic vulnerability of masonry structures and the effectiveness of sensors in detecting damage. The management of buildings in earthquake-prone areas can benefit from these specifications.

## 1. Introduction

In recent decades, Structural Health Monitoring (SHM) techniques have emerged as an indispensable tool for optimizing maintenance or detecting damage or material degradation in structures. The field of SHM is based on the integration of sensors, data acquisition systems, diagnostic tools, and numerical models [1,2,3]. By continuous and autonomous assessment of the static and dynamic behaviour, SHM systems can detect anomalies, track damage or, hopefully, predict potential failures before they occur [4]. This proactive approach not only enhances the safety and reliability of structures but also optimizes maintenance strategies, resulting in significant cost savings. In this framework, Operational Modal Analysis (OMA) has been established as a widely utilized technique for studying structures under environmental excitations, supporting applications such as Structural Health Monitoring, acceptance testing, and model updating [5]. In OMA, modal parameters are derived solely from data measured during environmental vibrations, which act as an unknown input (for instance, wind loads, micro-tremors, or traffic), without the need for artificially applied excitations on the structure. Moreover, OMA offers a modal model that reflects true operating conditions, including the actual boundary conditions, forces, and vibration levels. A key benefit of OMA is its non-intrusive nature, enabling its application during regular structure usage. OMA algorithms are developed on the assumption that input forces are stochastic, a common scenario for civil engineering structures such as buildings, towers, bridges, and offshore platforms, which are predominantly subjected to random environmental forces.

It is of the utmost importance that the SHM system used in the framework of OMA is able to be sensitive to low-level structural response and, among other devices constituting the system, sensors are the most important [1,2,3,5].

Accelerometers are commonly used to monitor changes in a structure’s natural frequencies due to damage or material degradation. For existing structures, OMA plays a crucial role in the structural knowledge acquisition process. In the case of monumental masonry buildings, the number of structural elements, the non-uniform distribution of mechanical parameters, and the orthotropic behaviour of the masonry can determine conditions in which a numerical model does not correctly reproduce the modal response of the building. This discrepancy can determine significant errors in the estimation of the seismic vulnerability of the structure. Thus, calibrating numerical models with observed signals using model updating techniques is critical [6,7]. For dynamic identification, ultra-low-noise seismic accelerometers, including analog piezoelectric sensors and those with built-in electronics for signal conditioning (IEPE), as well as force balance (FB) accelerometers, are commonly used for engineering purposes. Recently, the awareness of the importance of engaging OMA or SHM techniques for managing large and fragile infrastructures has encouraged private companies and researchers to explore the advantages of micro-electromechanical system (MEMS) accelerometers. In comparison to IEPE or FB accelerometers, MEMS accelerometers are advantageous in terms of their power efficiency and cost but not always in terms of resolution or noise floor, and, today, only a few products achieve high resolution and an ultra-low noise floor [8,9,10].

Although all the mentioned technologies are commonly used in SHM systems, a comparison of them on a real or to-scale structure has rarely been performed. Some authors [11] compared professional sensors with a wide range of low-cost MEMS sensors on a steel frame structure. They observed that the range, bandwidth, and cross-axis sensitivity of low-cost MEMS accelerometers can meet the requirements for SHM applications reported in the literature. However, their noise density is very far from the low-noise requirement in structural monitoring. Despite this limitation, they suggested that low-cost sensors might still be suitable for monitoring highly excitable structures, such as footbridges or bridges subjected to ambient excitation from winds or usage loads.

Aiming to investigate sensors’ performance on rigid structures, which represent most masonry structures, a scaled two-story masonry model subjected to seismic excitations was created. The benchmark, which was designed and built for dynamic testing on the shake table at the Experimental Dynamics Laboratory of the L.E.D.A. Research Centre, University of Enna “Kore”, was part of the PON project EWAS (Early Warning System for cultural heritage). The model replicates two-story masonry structures made by tuff regular blocks and mortar joints that represent the traditional construction technique used for the last three hundred years in Sicily, Italy, scaled to 2:3 using Froude similarity conditions [12]. This approach ensures the model accurately reflects the dynamic behaviour of such buildings under seismic conditions.

The testing campaign evaluated the dynamic behaviour of the scaled masonry model on a vibrating table and analyzed the performance of various types of sensors deployed. For this reason, the structure was instrumented with a combination of piezoelectric and MEMS accelerometers, selected for their sensitivities and self-noise densities.

Measurements were first taken under ambient noise conditions, followed by seismic tests of increasing intensity. The goal was to induce controlled damage states without causing collapse or irreversible damage to the model, enabling a detailed study of its dynamic properties and performance under progressive seismic loading. Comparing sensor performance before, during, and after each dynamic test provided data to assess their ability or not to identify structural parameters and low-level damage to the building under tests.

This manuscript is organized as follows: Section 2 describes the masonry specimen under testing, the shaking table facility, the characteristics and positioning of the deployed sensors, and, finally, the test sequence; Section 3 reports the results of the experimental campaign by focusing on the ability of the different sensors to identify the structural parameters; and, finally, Section 4 offers conclusions drawn from the study.

## 2. Materials and Methods

### 2.1. Building Characteristics

The structural system under testing was designed to emulate a typical two-story masonry building found in central Sicily (Italy), reflecting common floor heights, wall thicknesses, and sizes of openings. All dimensions were scaled down to a 2:3 length ratio to match the dimensions of the shaking table. The layout includes a main facade that is 2.96 m long, while the lateral walls extend 3.41 m. The total height of the structure reaches 4.49 m, with an approximate mass of 20 t. Constructed from tuff blocks measuring 16 cm × 25 cm × 40 cm and with 16 cm thick walls, the structure is bonded with M5 category premixed mortar.

The configuration of the openings (doors and windows with upper architraves) was designed to introduce significant eccentricity into the structural model, distinguishing the centre of gravity of the masses from that of the stiffness. Indeed, three walls feature off-centre openings on both levels, whereas the fourth wall is solid. Figure 1 details the model dimensions.

The foundation and floor curbs of the masonry building are composed by four concrete beams of class C25/30 at each level, equipped with longitudinal and transverse reinforcement bars of category B450C. The foundation beams have a cross-section of 25 cm × 40 cm, except for the beam under facade 1, which is an L-shaped cross-section in which the protruding part is equal to 25 cm with a thickness of 20 cm, while the stringer sections are equal to 16 cm × 20 cm on the first floor and 16 cm × 25 cm on the second floor. The structure was anchored using 13 M30 threaded bars made of class 8.8 steel, which connect the four foundation beams to the shake table. At each elevation, two balconies, made with metal profiles of section HE 100 B and class S275, are placed on facade 1 at both levels. The two floors were made of wood, as shown in Figure 2. Each floor is supported by five wooden beams with a cross-section of 10 cm × 13 cm of class C24, placed at approximately 75 cm apart from one other. These beams are covered by a double-frame board, with each frame having a thickness of 2 + 2 cm.

Under dynamic test conditions, additional masses were required to accurately account for the inertial forces acting on the real structure (prototype). These masses were added to compensate for the structural masses (G_1_) that were absent due to the similarity of the model adopted, and to consider the masses associated with an appropriate rate of permanent non-structural loads (G_2_) and variable loads (Q) with the appropriate load combination factor (Ψ_2_). Specifically, for practical and logistical reasons, a load equal to (G_2_ + Ψ_2_ Q) × A was considered in order to take into account actual values of loads according to local historical buildings and the Italian seismic code.

Table 1 reports the calculation of the total additional masses, along with those added during the experimental test for each of the floors and balconies. For the floors, these masses were created using loose material in bags (sand, cement, etc.) present and available in the laboratory. For the balconies, steel plates of appropriate weight were rigidly fixed to the metal beams of the balconies themselves.

### 2.2. Test Facility

To investigate the effectiveness of the proposed reinforcement technique during earthquakes, an extensive experimental campaign was carried out on the 3D shaking table system at the Experimental Dynamics Laboratory of the L.E.D.A. Research Centre [13,14]. This system consists of two identical 4 m × 4 m shaking tables, each with 6 degrees of freedom (DOF), which can operate either separately or simultaneously, and with asynchronous or synchronous motion (Figure 3a). Furthermore, the two tables can be connected by means of a rigid steel link exploiting a 10 m × 4 m 6-DOFs shaking table, which is currently the largest 6 DOF shaking table in Europe (Figure 3b). Table 2 summarizes the main characteristics of the two configurations.

The system’s high capacity allowed the building under test to be placed on a single shaking table. The building was connected to the shaking table using a reinforced concrete foundation designed to guarantee an adequate level of safety during both handling and experimental tests. A global view of the specimen is shown in Figure 4.

### 2.3. Instrumentation

This section describes the implemented monitoring systems, covering the type of sensors (piezoelectric and MEMS), channels, and positioning. On the structure, five different types of accelerometric sensors with various characteristics and self-noise, with different data acquisition devices, were deployed. Table 3 summarizes the main properties and characteristics of sensors typically engaged in SHM systems. The table, which does not aim to provide an exhaustive list of sensors, reports the principal characteristics of Digital 32-bit MEMSs (M-A352 manufactured by Seiko-Epson in Suwa, Japan), analog MEMSs (SI1003 and VS1002 manufactured by Safran-Colibrys in Yverdon-les-Bains, Switzerland), Digital 20-bit MEMSs (ADXL355 manufactured by Analog Device in Wilmington, DC, United States), Analog Piezoelectric (393B04, T333B50 and 356A17 manufactured by PCB Piezotronics in Depew, NY, United States) and force balance (Episensor ES-T by Kinemetrics in Pasadena, CA, United States). Those adopted in the experiment are identified by a code in the first column. In detail they are as follows: the Seiko-Epson M-A352 (a triaxial MEMS accelerometer (class A), which offers high sensitivity and ultra-low self-noise), the Safran-Colybyrs VS1002 (a single-axis accelerometer (class B), known for its low noise floor and wide dynamic range), the Analog Devices ADXL355 (a low-noise triaxial MEMS accelerometer (class B)), the two PCB Piezotronics T333B50 and the 356A17 (analog, triaxial piezoelectric accelerometers).

Figure 5 compares the Power Spectral Density (PSD) curves of the M-A352, VS1002, SI1003 (widely used in SHM), ADXL355 (widely adopted in commercial devices in the last decade), and Episensor ES-T (generally considered a standard in seismology and SHM) against low-noise model and high-noise model curves. The curves highlight the better performance of the M-A352, which closely matches that of the Episensor ES-T—an ultra-low self-noise force balance sensor widely used in seismological and SHM applications. Notably, despite being a MEMS accelerometer, the M-A352 exhibits a lower self-noise density compared to both competing MEMS sensors and an even better performance than the two piezoelectric accelerometers from PCB Piezotronics. The PCB 356A17 has a self-noise at 10 Hz of 6 µg/√Hz, comparable to those of the VS1002, whereas the PCB T333B50 has a slightly lower value of 3.8 µg/√Hz. The ADXL355 and VS1002 exhibit a significantly higher self-noise density compared to the others.

Due to the considerable number of sensors (34, corresponding to a total number of 92 seismic channels) a schematic representation of the specimen was needed for identifying the actual sensor placement. Table 4 reports the schematic view of the benchmark, numbering of the facades (F, numbered 1 to 4), the verticals (V, numbered 1 to 4), and the levels (L, numbered 1 to 2), and the global reference system. The scheme helps to identify the position of each sensor as reported in Table 4, where the sampling rates are additionally clarified. The table is coherent with the pictures reported in Figure 6.

### 2.4. Test Sequence

The campaign was designed with the aim of investigating the structural behaviour of the specimen during triaxial seismic inputs with increasing amplitude, up to the point where significant damage has been observed, while preventing the collapse of the building itself.

The adopted seismic excitation was the one that was recorded during the 2009 L’Aquila earthquake (M_w_ = 6.1), at the AQV station (L’Aquila Valle Aterno, belonging to the Italian Strong Motion Network and located 4.9 km from the epicentre). Here, the Peak Ground Acceleration, PGA, recorded along the NS, EW, and vertical components (X, Y, and Z directions of our global orientation system, respectively) were 0.550 g, 0.662 g, and 0.507 g, respectively. To standardize the seismic waveform, the three components of the original seismic input have been scaled to have a Zero Period Acceleration (ZPA) equal to 1 g for the EW component, which corresponds to the Y component of our global orientation system. These scaled components were then applied through five progressive load steps, ranging from 10% to 50% of the ZPA.

Moreover, the frequency content of the input signal was then varied iteratively, following one of the algorithms consolidated in the literature [15,16,17], in order to achieve correspondence with the response spectrum of Eurocode 8 for type C soil [18]. Thus, the derived signal retains the non-stationary characteristics of both time and frequency from the original natural signal, making it adaptable for a wide range of structural models. Finally, considering the geometric scale reduction applied to the model’s construction, the time axis was correspondingly reduced by a factor of $\sqrt{\left(2/3\right)}$. The resulting signal is depicted in Figure 7.

Before and after each seismic load increment, dynamic measurements were conducted in the presence of two different broadband noise excitations:Environmental noise when the shaking table was not operating.Three-direction broadband noise excitation by reproducing on the shaking table a three-component, uncorrelated Gaussian random noise in frequency ranges from 0.25 Hz to 60 Hz, with an RMS amplitude equal to 0.03 g.

Table 5 reports the testing sequence as conducted at the Laboratory of Experimental Dynamics (L.E.D.A.) on weeks 46 and 47 of 2023. The results of the environmental noise acquisition tests are not reported in Table 5, while “WN” indicates White Noise excitations and “EQ” stands for earthquake reproduction tests.

With the aim of comparing several accelerometer sensors and assessing their ability to identify structural parameters and low-level damage of the building under test, the described noise tests are also used to obtain the results discussed in the following sections.

## 3. Results

In the first phase of the study, we established baseline conditions through measurements under ambient noise. In the absence of any external seismic inputs, this phase was crucial for capturing the natural dynamic properties of the structure. By analyzing the recording of noise data in a ‘quiet state’, we established a reference point for the sensors’ behaviour and the modal frequencies of the structure. These baseline measurements provided a benchmark against which the subsequent seismic responses could be compared. The application of the Horizontal-to-Vertical Spectral Ratio (HVSR) and Operational Modal Analysis (OMA) are the two methods used for analyzing ambient noise. HVSR analysis helps identify the predominant modes of vibration. OMA, on the other hand, allows for the dynamic identification of structural properties and changes through the analysis of ambient vibrations over time. We preliminarily performed a comparative HVSR analysis on the average noise spectra recorded one month before the test (19 October 2024), during a period of relative quiet, with the shake table motors turned off, and during the test itself (22 November 2024) with the shake table motors in operation. The results showed that the vibrations induced by the shake table were localized in the lower frequency range (1.8–2.6 Hz) and did not significantly affect the overall environmental noise conditions for the OMA. The following paragraph discusses the results of day 22.

### 3.1. Analysis of Horizontal-to-Vertical Spectral Ratio (HVSR) of Ambient Noise

Horizontal-to-Vertical Spectral Ratio (HVSR) analysis is widely used by geophysicists to study the resonance frequency of sediments over bedrocks. Recent studies have also supported its application in structural engineering to evaluate the dynamic properties of buildings and their response to ambient noise, aiding in the assessment of structural integrity and seismic performance [19,20]. By analyzing ambient noise, the HVSR method can reveal the amplitude distribution and resonant frequencies across the floors of a building and provide an indicator of seismic activity effects.

In this study, HVSR analysis was applied to ambient noise data recorded during the shaking table tests (22 November 2024) by accelerometric stations equipped with Seiko-Epson M-A352 sensors (M1 to M9; see Table 4 and Figure 6). These sensors captured an extended ambient noise sequence before the seismic inputs. By analyzing these recordings, we aim to identify the natural frequencies of the structure under low-amplitude ambient conditions, providing insights into its vibrational characteristics. Moreover, this analysis offers a comprehensive understanding of how ambient vibrations propagate and interact with the structural system, and how the structure’s response may vary—or not—based on its geometry.

The HVSR analysis was performed using the Geopsy software v3.3.2 tool [21]. The ambient noise recordings used for the analysis have an average duration of over two hours, including a time frame of approximately 50 min before the startup of the shaking table test and an additional span of about two hours before the start time of the seismic input corresponding to 10% of the ZPA. Care was taken to ensure that no strong seismic disturbances were present in the time spans analyzed. Moreover, the signals that were not stationary were removed by applying the STA/LTA Anti-triggering Algorithm. The waveforms were analyzed using 60 s window intervals and in a frequency range of 0.1–30 Hz. More than 100 windows (130 on average) were used for the spectral analysis. In processing the HVSR curve, a smoothing process was performed using the Konno–Ohmachi method with a bandwidth coefficient of 40 and a width of 5%. In Figure 8, the resulting average HVSR curves at all nine accelerometric sensors are shown. All the sensors are installed on levels 1 or 2 of the structure and on the external facades, except for sensor M8, which is installed inside the structure at ground level (level 0), next to the internal part of facade 3 (Figure 6). For a given facade, the HVSR curves are almost completely overlapping, differing only in amplitude. In particular, the accelerometers located along verticals V3 and V4 (see the graphical representation inside Table 4), installed at the second level (M4 and M9, respectively, along V3 and V4), exhibit more amplified HVSR curves compared to those installed along verticals V1 and V2 (M5 and M2, respectively). However, although slightly different, these curves exhibit significant similarities in the resonance frequency values. Indeed, both sensors show a broad peak overall in the frequency range between approximately 2 and 3 Hz (with peaks at 2.2 Hz and 2.5 Hz, respectively, for facades 3 and 1), which can be ascribed mainly to the dynamic response of the entire system of the shake table. These peaks at low frequencies are also particularly evident at sensors M1 and M3, which are installed at the first level, along verticals V3 and V4, respectively. Other peaks are evident at higher frequencies, particularly at around 7.7 Hz and 17 Hz in both groups of HVSR curves. These peaks can be attributed to the vibration modes of the structure and are generally more pronounced at all sensors on the second level of the structure (M2, M4, M5, and M9).

This increase highlights a growing difference in acceleration between levels, which amplifies the structure’s vulnerability at those resonance frequencies.

### 3.2. Structural Dynamic Behaviour

Before elaborating on the signals, a preliminary estimation of the fundamental frequency was carried out by means of simplified formulas available in both the literature [22] and technical regulations [18,23,24].

Given that the specimen height (H) is 4.09 m and the dimensions (D) are 2.96 m × 3.41 m, Table 6 presents the correlation between the fundamental frequency and geometric properties. As discussed in other authors’ works [22], several modifications of the simplified formulas suggested by the design codes [18,23,24] have been proposed in the literature, aiming to minimize errors observed in measurements on existing structures. Notably, two relationships proposed by Alguhane et al. [22] stand out. As commented by the author himself, the formulation proposals can be interpreted as two upper and lower limits within which the frequency parameter should be placed. Although numerous literature studies have pointed out the unreliability of the formulas established in design codes, it has to be considered that the structure under examination is not a full-scale building, and further differences could be inherent even for more calibrated formulas such as those adopted in this section. According to Table 6, a fundamental frequency may be expected between 6.01 Hz and 7.49 Hz at an average value of 6.54 Hz.

As described in the previous sections, the sensors acquired accelerations before, during, and after each seismic input and a preliminary test was carried out with ambient noise in which all the systems acquired data under extremely low levels of ambient accelerations. All the data have been elaborated by using the SSI-UPC [25] algorithm implemented in ARTeMIS 7.2 software [26]. The analysis aimed to reproduce in situ conditions generally present during OMA tests. The test results are summarized in Figure 9.

The analysis exclusively focuses on the X and Y channels, deliberately excluding the vertical channel to mitigate potential disturbances associated with it. The rationale for excluding the vertical channel is based on extensive preliminary testing, which indicated that vertical measurements often introduce noise and artifacts that could skew the interpretation of the data.

The findings are systematically categorized to provide a comprehensive understanding of the dynamic behaviour observed during the specified periods. By concentrating on the horizontal channels, the analysis ensures a clearer, more accurate depiction of the dynamic characteristics of the to-scale structure. As Figure 9 shows, the 0.2 µg/√Hz noise density sensors identified the first structural frequency at 6.985 Hz, and the 18-6-2 µg/√Hz @ 1-10-100 Hz noise density sensors identified the first structural frequency at 6.872 Hz; the others were not able to identify the same frequency. Even if the frequencies are coherent with the expected values (6.96 Hz), they were obtained under a certain level of additional ambient noise (0.03 g) due to the activation of the hydraulic pumps.

Aiming to evaluate the limits of these sensors to real-life SHM applications on masonry structures, the data acquired when the pumps were switched off have been analyzed. In this condition, only sensors with 0.2 µg/√Hz noise density were able to acquire reliable data for the OMA analyses. In Figure 10, the OMA analyses identify the first frequency at 7.691 Hz, which is 10% higher than the frequency identified under the hydraulic pump noise. It is worth noting that the 7.691 Hz frequency was identified during an acquisition on 19 October 2024 with an external temperature of 22 °C, while the 6.985 Hz frequency was identified on 22 November 2024, with an external temperature of 13 °C. The two values also might have been affected by the different weather conditions. The stabilization diagrams are elaborated on for all the sensors. Figure 9 and Figure 10 show the results obtained with the SSI-UPC algorithm where the singular value decomposition curves are coloured in blue (first singular value), red (second singular value), green (third singular value), and grey (all the others). All the records were processed using the same configuration, applying a decimation down to 25 Hz, a spectral density estimation with a resolution of 512 Hz, and an overlap of 66%. The SSI was set with a maximum state space dimension of 100.

As Figure 9 shows, all the sensors identify frequencies but not all of them are the same or can be considered reliable with the aim of a dynamic identification. Only the M-A352 (0.2 µg/√Hz) and PCB356A17 (18-6-2 µg/√Hz @ 1-10-100 Hz) sensors returned reliable data. Due to the poor capacity of identifying reliable modes, the sensors with VS1002 (7 µg/√Hz) and ADXL355 (25 µg/√Hz) are not compared in the following. The modal shapes and the frequencies are compared in Table 7. The frequencies and the modal shapes calculated before and during the hydraulic pumps’ activation can be considered reliable and stable (Table 8) for M-A352. Although the frequencies associated with the principal modes may appear similar, a significant discrepancy between the modal shapes is notable during the hydraulic pumps’ activation between M-A352 and PCB356A17 (Table 9).

Due to the short duration of the signals and specific laboratory testing conditions, which introduced numerous harmonic components from active instruments during the tests, many values exhibit a high complexity index, often associated with data inaccuracies. Additionally, some tests failed to identify the frequency of interest. However, it is observed that, during the test, the frequency value decreases, likely due to the progressive damage of the model, as shown in Table 10. In the table, the levels of acceleration (from 10% to 50% of the ZPA) divide each chunk of data from another.

It is noteworthy that the modal shapes associated with the first two significant frequencies are flexural, as observed in the 01_R1345_1355 recording. At the end of the test, the modal shapes retain the same orientation but show significant reductions in frequency (from 6.98 Hz to 5.014 Hz for the first mode). Comparing the modal shapes of the first and last recordings, a noticeable irregularity is observed between the translational components of the two points aligned at the first level, which could be linked to the localization of damage at this level, as shown in Figure 11.

The frequency variations due to the progressive damage to the structure are confirmed in Figure 12, which presents an example of an analysis of the seismic inputs at the 10%, 30%, and 50% ZPA intensity levels (Figure 12a), performed using both the non-normalized and normalized Stockwell transform. The data were recorded by the M5 accelerometer located on the second level of facade 1. The Stockwell transform (S-transform) is a hybrid method combining the features of the Short-Time Fourier transform (STFT) and wavelet transform [27], providing a localized frequency spectrum. Among time–frequency techniques, the S-transform allows for the extraction of processed information from the signal without altering its local spectral characteristics. It has been successfully applied in various civil engineering applications and in Structural Health Monitoring [28,29].

The standard and non-normalized S-transform uses a Gaussian window with a fixed amplitude, constant across all frequencies. While this approach is effective for analyzing the energy distribution, normally, it does not point out relative amplitude variations across frequencies. Figure 12b, however, highlights significant differences between the scalograms related to 10%, 30%, and 50% of the ZPA. In particular, during the seismic inputs, the scalograms of both the X and Y components exhibited a progressive increase in energy at lower frequencies. Conversely, the Z component manifests a progressive increase in energy at higher frequencies.

To improve the comparability of amplitudes across different frequencies, we also performed a normalized analysis (Figure 12c) by introducing a frequency-dependent scaling factor, which ensures a consistent amplitude of the Gaussian window across frequencies. This adjustment enhances the sensitivity of the analysis to variations in the signal amplitude and frequency content. Considering the varying signal levels, a dynamic computation of the normalized S-transform was necessary, proving crucial for analyzing noise signals both before and after the seismic inputs.

In Figure 12c, 60 s of signal containing seismic inputs at 10%, 30%, and 50% of the ZPA were dynamically normalized. This normalization made it possible to detect subtle shifts in noise characteristics caused by progressive structural damage. In addition to the significant frequency variations observed during the seismic recordings, the normalized scalograms revealed changes in the frequencies of the structure related to its dynamic behaviour outside the period of the seismic inputs. Overall, we observed a downward shift of approximately 0.2 to 0.5 Hz in the post-seismic noise frequencies across all of the sensor components compared to the pre-seismic noise (Figure 12c). However, after a certain period, these frequency values partially recover, returning to their pre-seismic levels, as we note in subsequent seismic inputs.

These intriguing yet preliminary results from the application of the S-transform highlight the need for further in-depth analysis of the signals recorded by the various sensors. At present, an extensive analysis lies beyond the scope of this work.

### 3.3. Strong Motion Parameters

During the experimental test conducted on the shaking table, we analyzed the motion peaks, focusing on two key parameters: the peak acceleration, both at the ground level (Peak Ground Acceleration, PGA) and at the two floor levels (Peak Floor Acceleration, PFA), and the Pseudo-Spectral Acceleration (PSA). These parameters are essential for evaluating the structural response of the construction during the several phases of the test, which involve a progressively increasing seismic input.

Similarly to what was carried out for the calculation of the HVSR curves, the data used for the estimation of PGAs, PFAs, and PSAs are still those collected by the stations from M1 to M9, which, as a reminder, are equipped with the Seiko-Epson M-A352 sensors. Additionally, for comparison purposes, we also considered all the stations installed next to vertical 1 (V1; see the graphical representation in Table 4), namely, the A2 and A6 stations (equipped with PCB Piezotronics T333B50 sensors and co-located at stations M6 and M5, respectively), and the M11 station (equipped with an ADXL355 sensor) (see Figure 6a,b for their installation locations on the structure).

Figure 13 reports the PFAs and PSAs measured at the aforementioned stations along the three directions of motion, normalized with respect to the ground-level PGA and PSA, which are measured at station M8 (PGAM8 and PSAM8, respectively), the only station installed at ground level. The normalized PFAs and PSAs show a clear amplification as we move from the first to the second level of the structure, especially along the horizontal component of motion and across all the seismic input intensities. Conversely, along the vertical component of motion, the normalized PGAs and PSAs exhibit a much more complex and irregular trend, with no clear tendency to either increase or decrease from level 1 to level 2. Moreover, as the shaking intensity increases, reaching and exceeding 40% of ZPA, the differences between the trends observed at individual stations become more pronounced, especially for the stations installed along the vertical V1 of the structure and at both levels 1 (M6, M11, and A2) and 2 (M5 and A6). In particular, along the vertical component, Figure 13 shows a clear deviation from linear behaviour once the seismic input reaches and exceeds 40% of the ZPA. The threshold of 40% seems to mark a significant change in the structural response, indicating that the structure undergoes a transition to a non-linear behaviour as the intensity of the seismic input increases. This behaviour is particularly evident at the sensors installed in the part of the structure that has suffered greater damage due to the formation of impact-induced fractures right next to the sensors M3, M5, M11, A2, and A6 (possible rocking effect). This suggests that the structure’s ability to maintain a proportional response to seismic forces diminishes beyond this point, highlighting potential issues with the stability or performance under higher seismic loads.

Still focusing on the sensors installed along the vertical V1, we consider the response spectra for the various increasing percentages of the ZPA, as computed for the sensors installed at levels 1 and 2. As is easy to notice, the spectral shapes indicate a strong frequency-dependent behaviour of the structure, with high peaks suggesting significant resonant behaviour, potentially indicating structural vulnerability. For a given component, the values of the peak spectral accelerations (PSAs) increase as the seismic input increases, as expected. Similarly, for a given component, the PSA values increase from the first to the second level of the structure, as equally expected. Moreover, there are no significant differences in both the spectral shape and PSA across the different sensors, which exhibit a consistent response. An exception is the vertical component of sensor M11, which shows an ‘anomalous’ behaviour once 50% of the ZPA is reached, with an increase in long-period components (i.e., low frequency) and a PSA reaching 0.216 s (4.6 Hz). The anomalous trend observed at this sensor for 50% of the ZPA highlights what was previously shown in Figure 13a (normalized PFA and PSA values that strongly deviate from linearity) and could be attributed to impact effects resulting from the formation of significant fractures right at this sensor, which open as soon as the seismic input reaches 50% of the ZPA. From the comparison of the PSA values calculated for the different inputs, it emerges that the Y component is scarcely influenced by an increase in the seismic input, with values ranging from 0.245 s to 0.26 s (3.8–4.1 Hz). A similar trend is observed along with the X component, at least until the seismic input reaches 30% of the ZPA, beyond which it tends to show values that fluctuate between lower and higher. Along with this component, the PSA periods globally range from 0.15 s to 0.37 s (2.7–6.7 Hz). Finally, along the Z component, the PSA period values are generally lower when compared to those estimated for the X and Y components, and this is true for both levels of the structure, with values ranging from 0.035 s to 0.225 s (4.4–28.6 Hz).

Normalizing the response spectra by the PGA at ground level (PGA measured at sensor M8) removes the effect of earthquake intensity, which is represented by the percentage of ZPA, highlighting the structure’s dynamic behaviour. As shown in Figure 14, the response spectra normalized to the PGA still suggests a different response of the two levels of the structure to the seismic excitation, particularly along the horizontal directions, relative to the ground motion. Indeed, the second level of the structure amplifies motion with respect to the ground more than the first one.

Along the X and Y directions, the structure’s levels experienced, on average, acceleration greater than the ground for periods shorter than about 0.8 s (frequencies higher than 1.25 Hz). The period decreases to 0.5 s (frequency higher than 2 Hz) along the Z-direction.

The analyses presented in Figure 13, Figure 14 and Figure 15 indicate that all of the sensors used exhibit consistent behaviour under seismic excitation. However, differences emerge as the input excitation increases beyond 40% of the ZPA, with the most significant deviations occurring primarily in the vertical (Z) direction. These findings will be further explored in future studies, where the detected damage pattern (extension and position) will be considered as a potential factor contributing to these discrepancies.

### 3.4. Damage Distribution

In line with the aims of the shake table test, the specimen was progressively damaged. Figure 16 reports the damaging pattern for each facade. The cracks are represented by coloured lines for each of the seismic input steps. As the data already identified, the more extensive damage appears at 40% and 50% of the ZPA of the target input. All the cracks follow the mortar layers. The ground level has been heavily damaged compared to the second level. Facade 3 and 4 are mainly denoted by shear failures. The cracks on facade 1 and 2 appear consistent with in-plain flexural failures. As a matter of fact, a few cracks were present before starting the test; they are shown in grey in the figures (Facade 1). A detailed view of two damaged areas is presented in Figure 17.

The sensors, which are summarized in Table 11, have been compared in terms of seismic response. In the following, the sensors that belong to the same vertical (V1) are compared. During each seismic input, the accelerations are considered in their by-axial components. The results are compared in Figure 18.

## 4. Conclusions

In this paper, the results of an experimental campaign carried out at the Experimental Dynamics Laboratory of the L.E.D.A. Research Institute of the University of Enna “Kore” on a 2:3 scale two-level masonry structure is reported. The specimen has been investigated using several accelerometric sensors for Structural Health Monitoring (SHM) purposes.

The main aim of our work was to evaluate the advantages and disadvantages of these sensors, focusing on some critical characteristics such as sensitivity and self-noise. Then, the performance of a variety of accelerometer sensors—namely, Seiko-Epson M-A352, Safran-Colibrys VS1002, Analog Devices ADXL355, PCB Piezotronics T333B50, 393B04, and 356A17—has been compared in their application to a scaled real structure subjected to both ambient and seismic excitations. Due to the reduced value of the scale factor (2:3), the results can be considered valid for both the actual and scaled building since their structural main frequencies, although higher in the scaled specimen, are still in the working range of the used sensors.

It is important to note, however, that evaluating the accuracy of sensors in capturing the system’s response under normal operating conditions and seismic events is just one critical aspect to consider in SHM. The success of SHM systems in real structures relies not only on the careful selection of sensors but also on their proper installation, effective data transmission methods, and efficient data processing strategies, beyond device and sensor maintenance, data volume management, and processing time.

To summarize the main outcomes of this work, the following points highlight the key conclusions:All the MEMS sensors used during the experiment demonstrated their ability to accurately measure the dynamic response during seismic excitation. Instead, their suitability for Operational Modal Analysis (OMA) under noisy conditions, such as during the experiment, varies significantly due to their self-noise characteristics.Overall, also for the piezoelectric sensors, the self-noise could become a crucial factor in very-low-ambient-noise environments and for OMA applications, affecting the quality of the results. Therefore, we retain that both piezoelectric and MEMS (micro-electromechanical systems) accelerometers with self-noise levels below 1 µg/√Hz—or preferably below 0.5 µg/√Hz—are recommended for effective OMA in environments with very low noise. In general, ultra-low sensor self-noise minimizes the interference from environmental noise, allowing for the more precise identification of modal frequencies and damage detection.For MEMS sensors with self-noise greater than 1 µg/√Hz, their performance, as mentioned above, in very-low-noise conditions may be significantly compromised and mask the acceleration signal. Additionally, other noise sources like wiring and electronic components can further degrade the measurement quality. However, in SHM applications in environments with high noise levels, such as busy bridges, footbridges, or civil structures exposed to strong winds, these sensors may still provide adequate performance.The OMA identified a frequency decrement during the tests. The extensive OMA campaign and the comparison between several sensors confirmed the necessity to engage sensors with a low (or ultra-low) noise density if the frequency identification is needed, both before and after a seismic event for real applications. A suggested value of less than 1 µg/√Hz—or preferably below 0.5 µg/√Hz—may be appropriate for the analysis of stiff masonry structures. This aspect is crucial if a reliable estimation of damage has to be carried out after an event under ambient noise conditions.The results obtained from the Stockwell transform are coherent with the expected dynamic behaviour of the structure and the impact of seismic loading. Indeed, a shift in lateral modes to lower frequencies can indicate reduced stiffness in the horizontal direction, whereas increased energy at higher frequencies in the Z component could be attributed to vertical accelerations and structural rigidity in the vertical direction. It also indicates localized damage affecting the sensor’s response to seismic inputs, as higher frequencies are generated by local and global rocking of the masonry structure.The acceleration response maintains the same proportion among the components for the first, second, and third loading steps. This aspect can be interpreted as an elastic response condition of the structure under incremental loads. From this step onwards, and more evidently in the subsequent step, the maximum recorded acceleration shows significant variations in orientation. This phenomenon may be related to the disassembly of the masonry structure due to damage. This is consistent with the conditions observed during the experimental test, where cracking patterns emerged from the fourth loading step. The crack pattern survey, conducted after each experimental test and reported in Figure 16, illustrates that at the fourth step (40% of the load), large cracks corresponding to the vertical alignment of station M5 are evident. In the subsequent step (50%), a widespread crack pattern significantly affects the in-plane resistance of the panel and, in general, the masonry piers. This results in higher out-of-plane accelerations.It is evident that the amplification of motion with respect to the ground decreases as the intensity of the seismic input increases, suggesting a progressive change in the structure’s damping behaviour. The height and sharpness of the peaks in the normalized response spectra decrease, suggesting a change in the structure’s ability to dissipate seismic energy, potentially indicating higher damping.The eccentricity between mass and stiffness centres due to the presence of solid walls (facade 3) led to an unequally distributed damage pattern. This response closely mirrors real cases, where structural irregularities lead to the damage of the structural elements with different mechanisms. As a result, the shear and flexural failures of the masonry walls define the damage at 50% of the target input level. The damage significantly affected the specimen’s dynamic behaviour, as identified by most of the sensors. It is worth noting that some differences have been identified in the acceleration peaks captured by each sensor type.

As a future development of this work, a numerical model will be investigated and updated to properly match the experimental dynamic behaviour of the masonry structure under testing. The updated model will be used for damage identification, a structural vulnerability assessment, and to guide the retrofit design process.

## Figures and Tables

**Figure 1 sensors-25-01010-f001:**
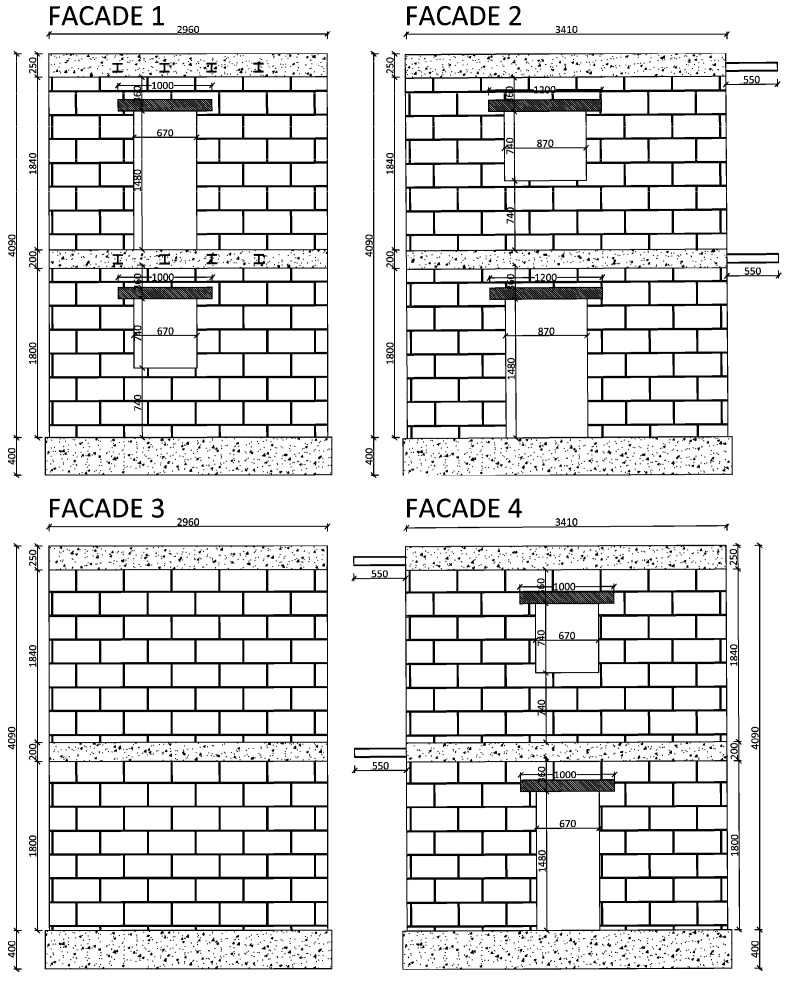
Design dimensions of the four facades of the specimen (units: mm).

**Figure 2 sensors-25-01010-f002:**
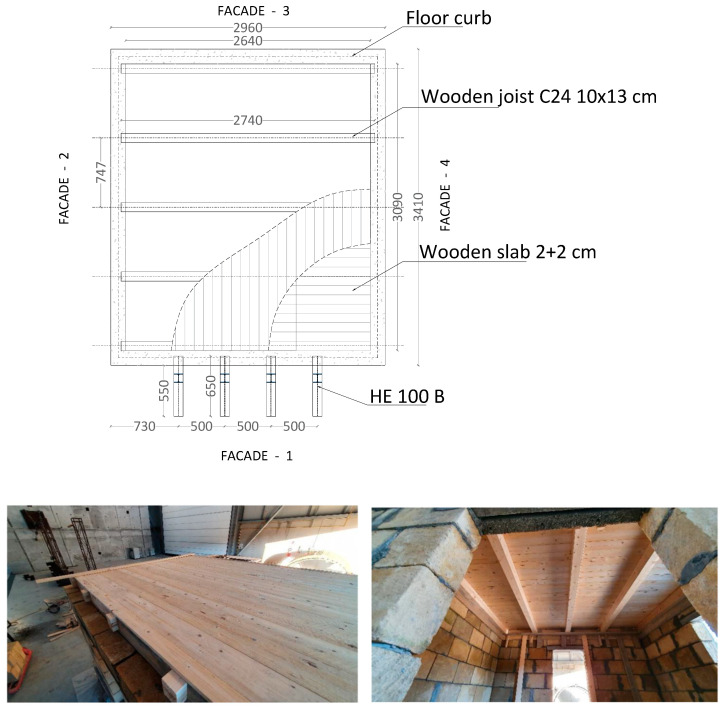
Plan view (units: mm) and construction of the wooden floors.

**Figure 3 sensors-25-01010-f003:**
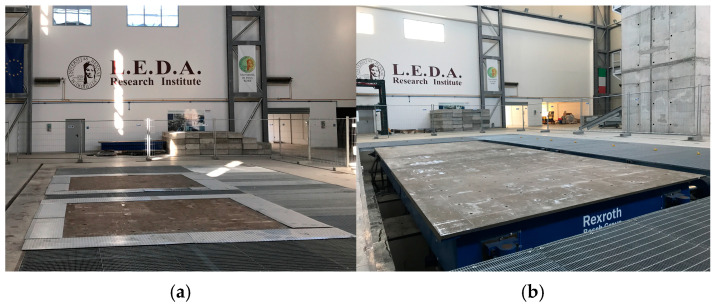
Shaking table system at L.E.D.A. Research Institute: (**a**) single tables; (**b**) connected tables.

**Figure 4 sensors-25-01010-f004:**
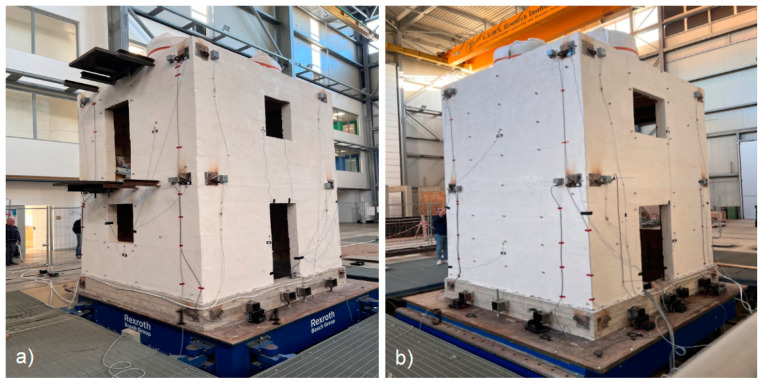
Specimen under test placed on the shaking table. From left to right: (**a**) facades 1 and 4; (**b**) facades 3 and 2.

**Figure 5 sensors-25-01010-f005:**
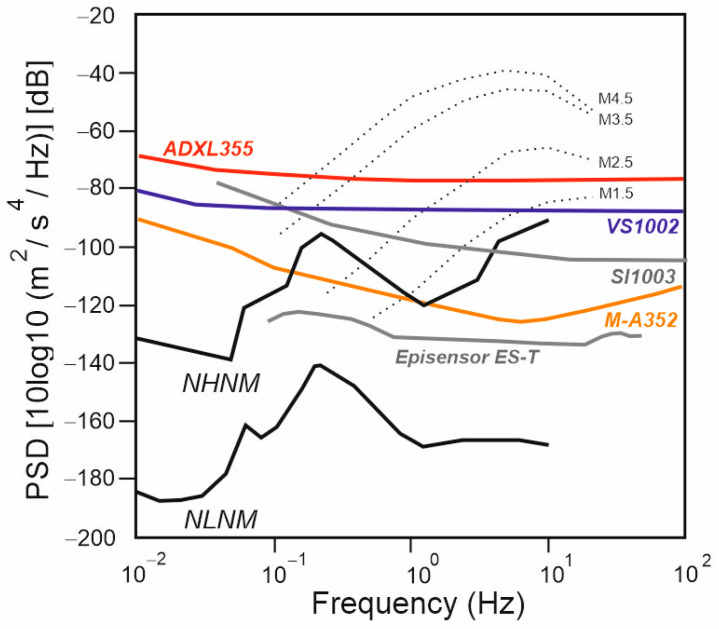
Comparison of the power spectral densities (PSDs) of self-noise for the three MEMS accelerometers used: Analog Device ADXL355 (red line), Safran-Colibrys VS1002 (blue line), and Seiko-Epson M-A352 QMEMS (orange line). The PSDs of self-noises for the Safran-Colibrys SI1003 (grey line) and of the Episensor ES-T force balance accelerometer (grey line), commonly used for seismology and SHM measurements, are also shown. Lastly, the seismic low-noise model and seismic high-noise model curves (thick black lines) are shown, along with the spectra of earthquakes of different sizes that were measured 10 km from the epicentre (point lines) (modified after Patanè et al. 2024 [2]).

**Figure 6 sensors-25-01010-f006:**
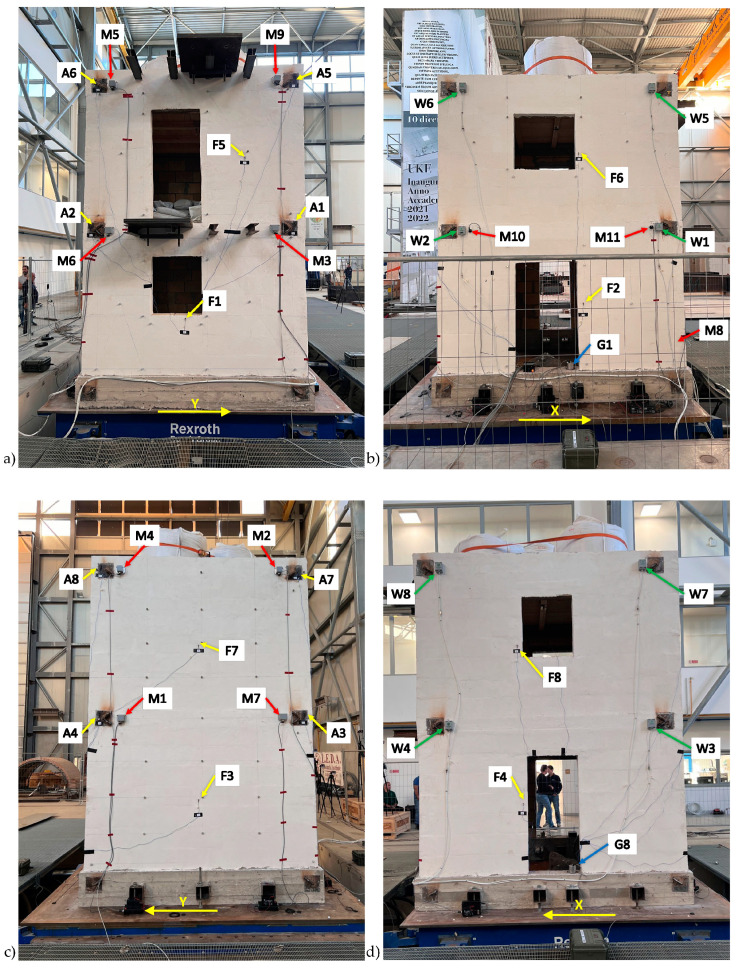
Distribution of the sensors installed on the individual facades of the building under testing: (**a**) facade 1; (**b**) facade 2; (**c**) facade 3; (**d**) facade 4. Refer to Table 4 for the sensor codes.

**Figure 7 sensors-25-01010-f007:**
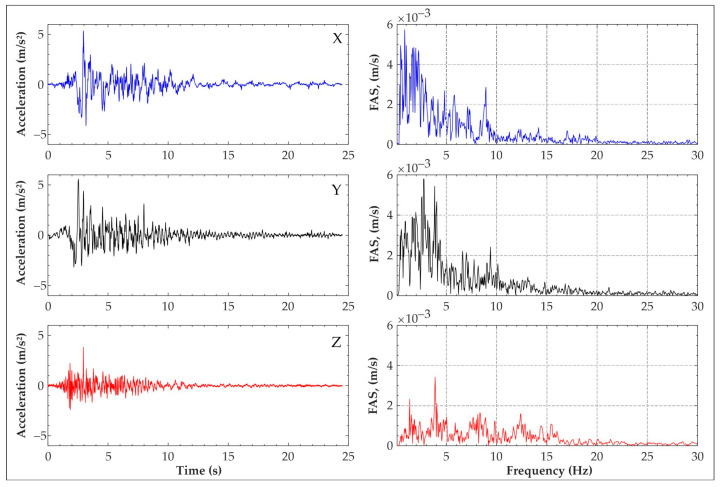
Scaled reference accelerograms for a seismic input at 50% of ZPA and the associated Fourier Amplitude Spectra (FAS).

**Figure 8 sensors-25-01010-f008:**
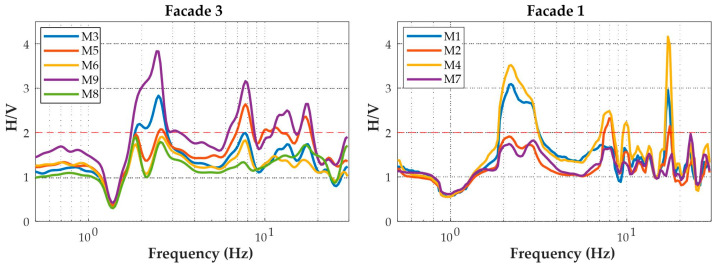
Average HVSR curves at the nine accelerometric stations equipped with Seiko-Epson M-A352 sensors.

**Figure 9 sensors-25-01010-f009:**
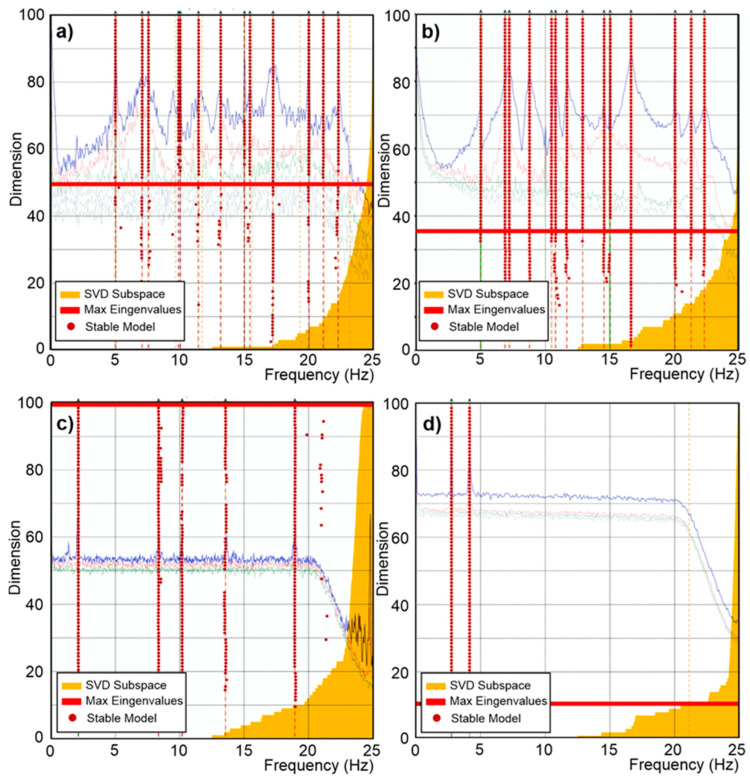
Stabilization diagram of estimation state space models of sensors with (**a**) 0.2 µg/√Hz, (**b**) 18-6-2 µg/√Hz @ 1-10-100 Hz, (**c**) 7 µg/√Hz, and (**d**) 25 µg/√Hz during the hydraulic pumps’ activation.

**Figure 10 sensors-25-01010-f010:**
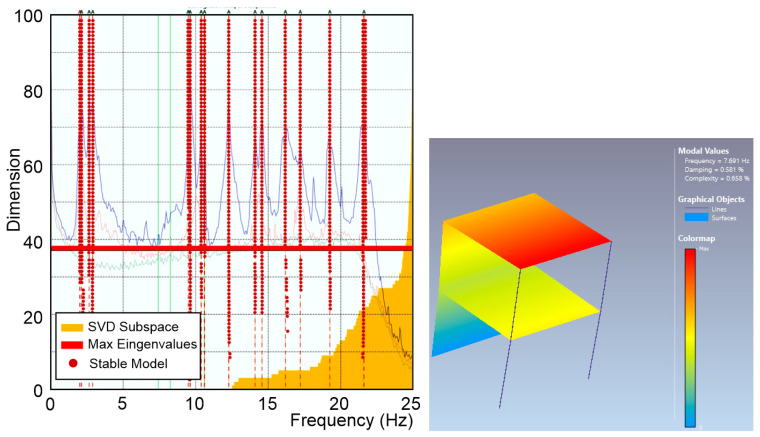
Stabilization diagram of estimation state space models of sensors with 0.2 µg/√Hz and modal shape.

**Figure 11 sensors-25-01010-f011:**
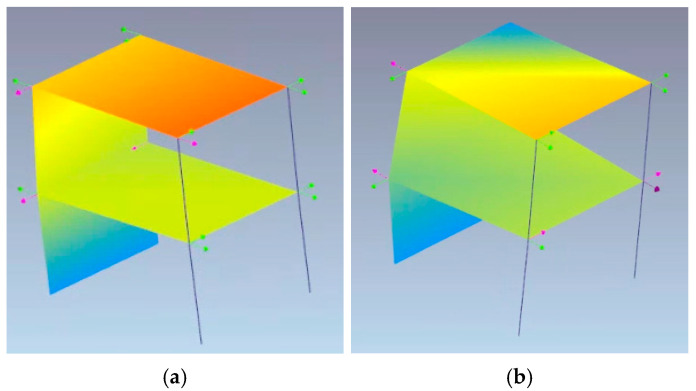
Comparison of the first frequency (**a**) before shaking test, and (**b**) at the end of shaking test.

**Figure 12 sensors-25-01010-f012:**
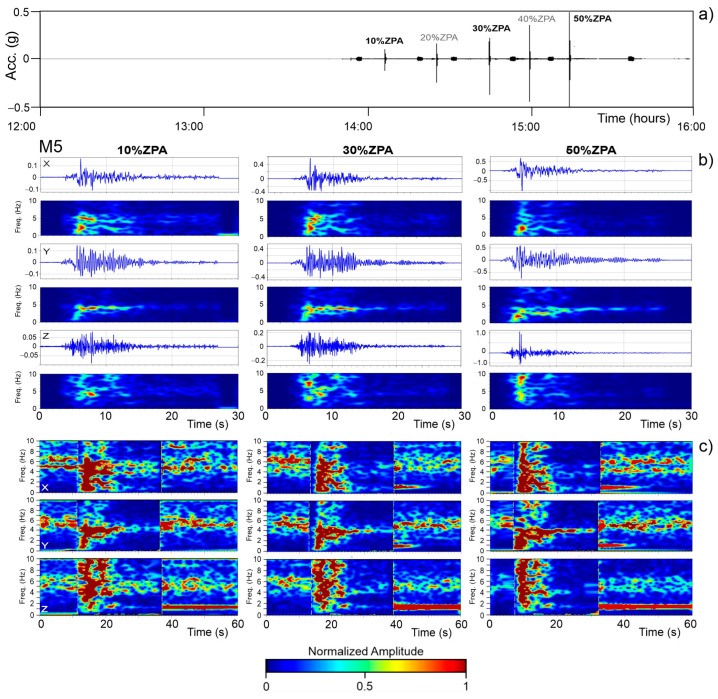
Seismic signal recorded during the experiment with different inputs from 10% to 50% (**a**). Stockwell transform, non-normalized (**b**) and normalized (**c**), for the signal inputs of 10%, 30%, and 50% recorded at accelerometer M5.

**Figure 13 sensors-25-01010-f013:**
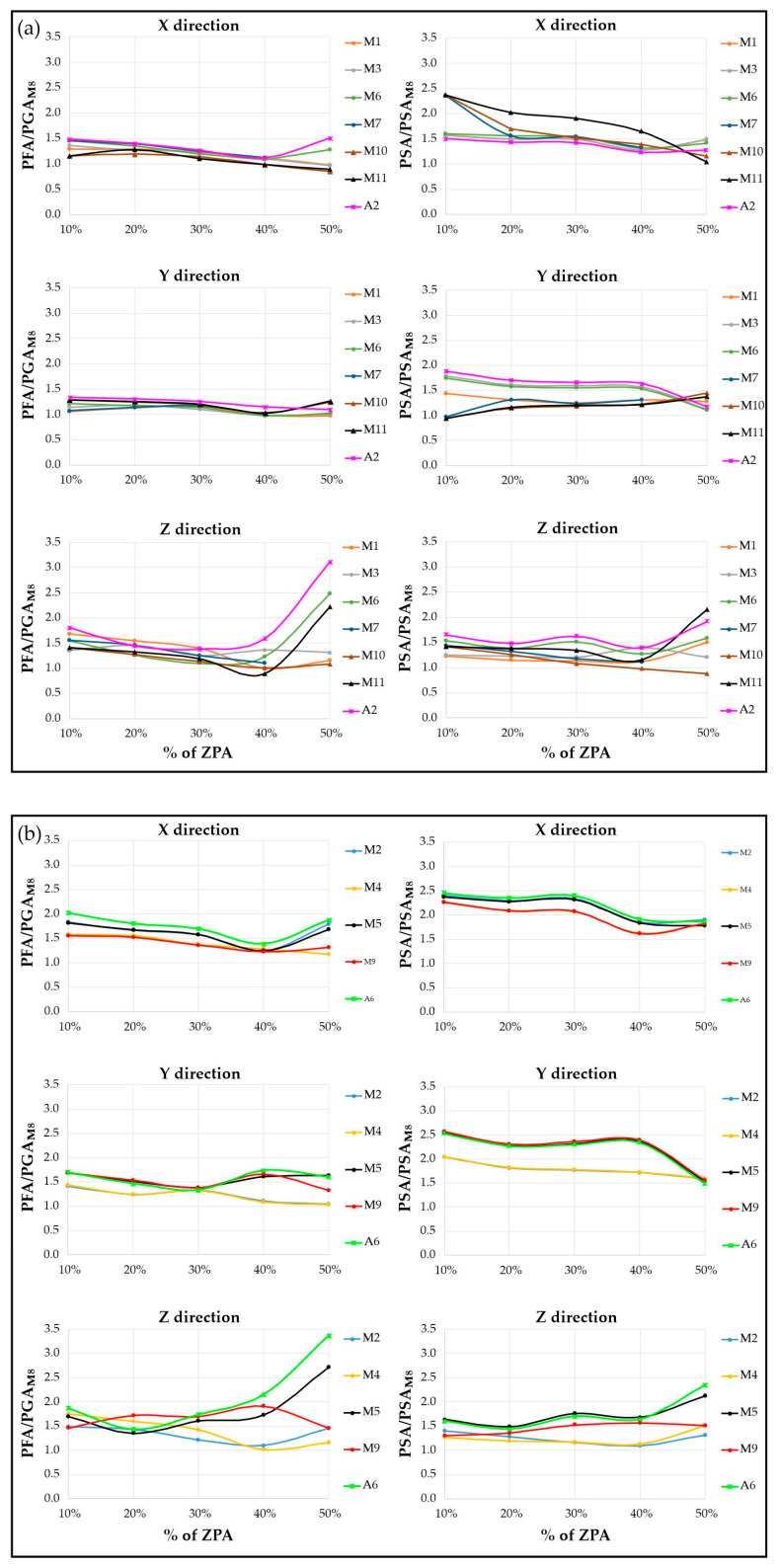
PFAs and PSAs measured at the stations installed at the first (**a**) and second (**b**) levels of the structure, normalized with respect to the ground-level PGAs (measured at station M8), for the different percentages of ZPA (%g) experienced during the shaking table test.

**Figure 14 sensors-25-01010-f014:**
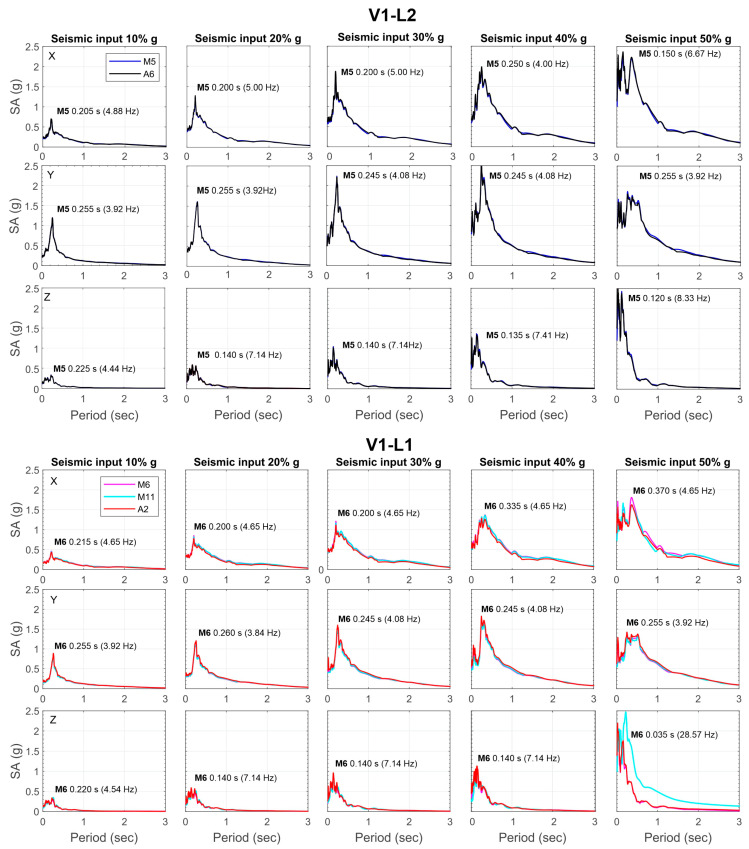
Elastic response spectra calculated for increasing seismic input at the stations installed along the vertical V1 of the structure and for the three directions of motion. In each plot, the values of the period corresponding to the PSA at stations M5 and M6 (arbitrarily chosen as the reference for levels 2 and 1, respectively) are also shown.

**Figure 15 sensors-25-01010-f015:**
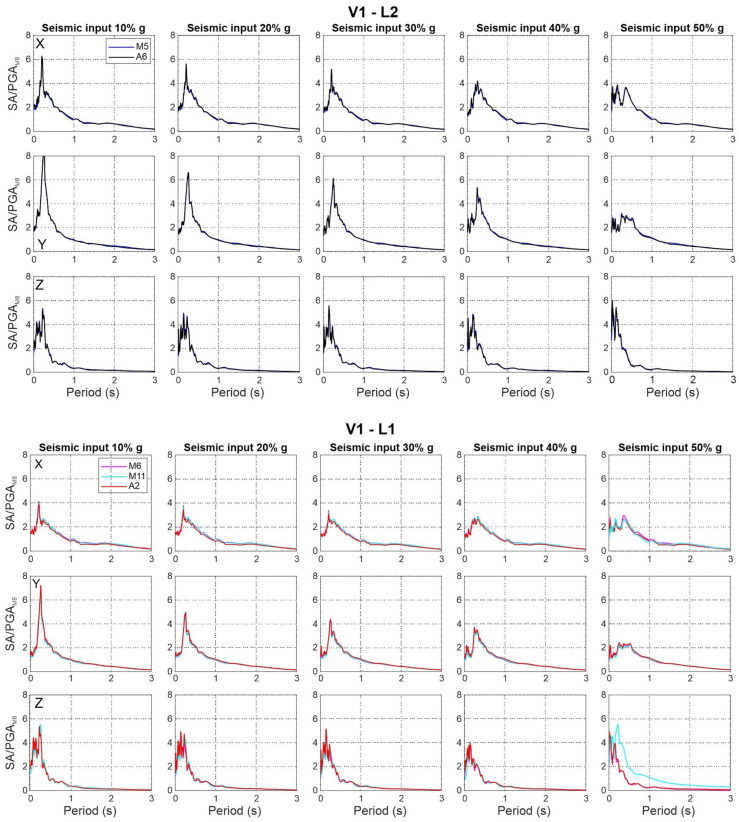
Normalized response spectra with respect to the values measured at the M8 station installed at the ground level.

**Figure 16 sensors-25-01010-f016:**
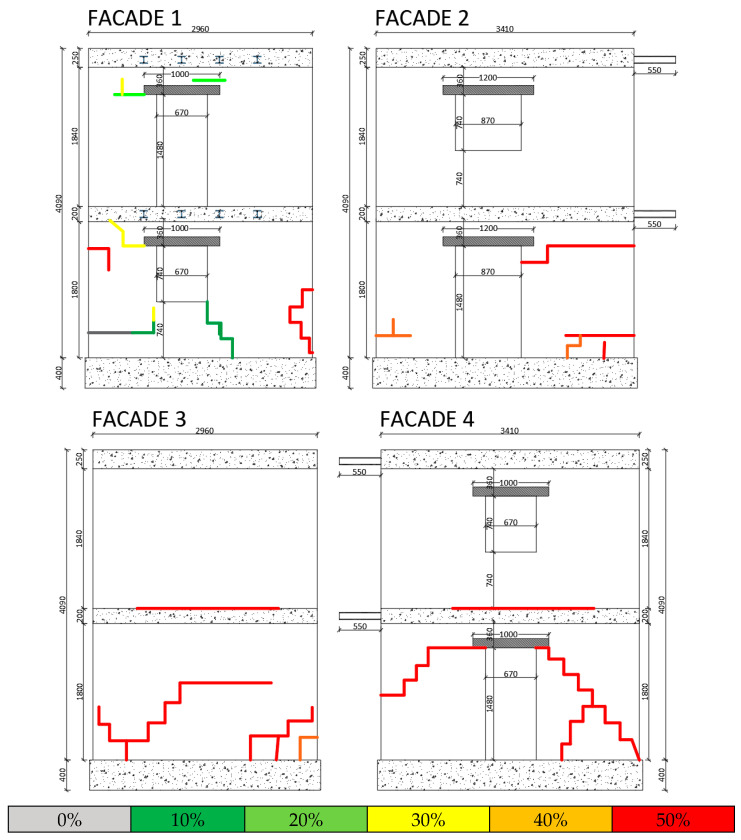
Qualitative damage distribution on all four facades.

**Figure 17 sensors-25-01010-f017:**
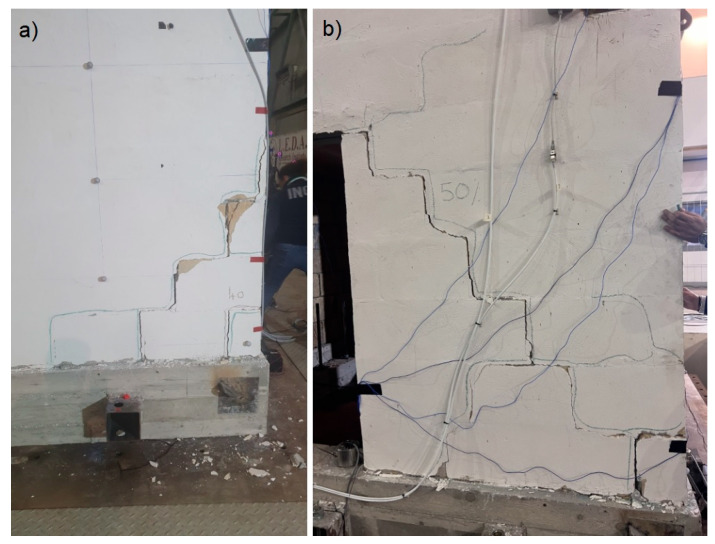
Details of two damaged areas: (**a**) the lower right corner of facade 3, and (**b**) the right masonry wall of facade 4.

**Figure 18 sensors-25-01010-f018:**
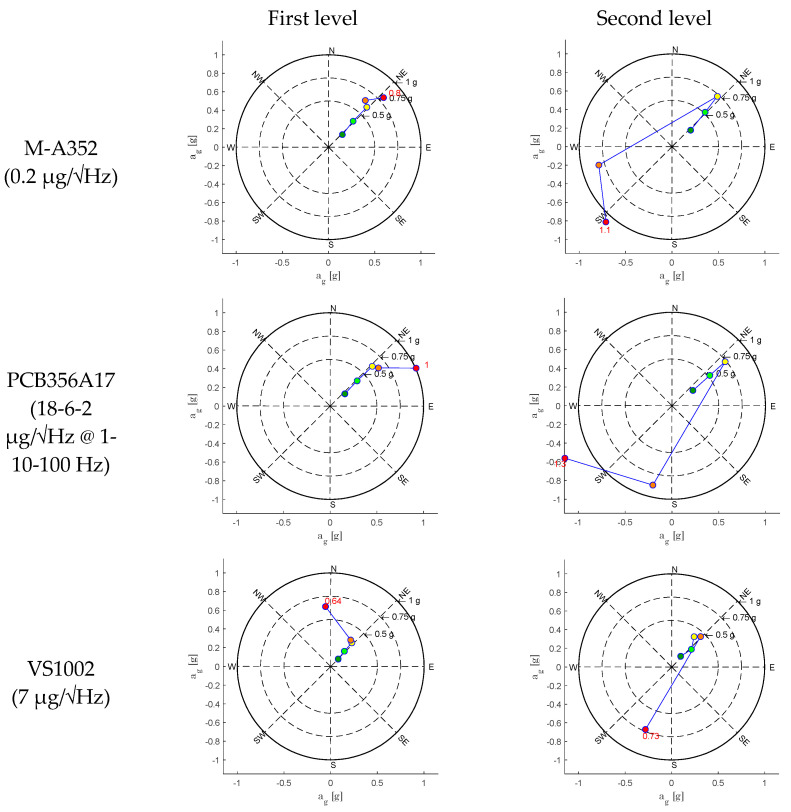
Planar seismic response for each sensor type on vertical V1 at each level.

**Table 1 sensors-25-01010-t001:** Additional masses.

Masses	Floor 1 [kg]	Floor 2 [kg]	Balconies [kg]
G_1_ prototype	10,659	7223	65
G_1_ model	7106	4815	43
Additional mass G_1_	3553	2408	22
(G_2_ + Ψ_2_Q) × A	1192	746	191
Total additional mass	4745	3154	213
Actual additional mass	4809	3150	221

**Table 2 sensors-25-01010-t002:** Dynamic properties of the shaking table system at L.E.D.A. Research Centre.

System Characteristics	Single Tables (Each)	Connected Tables
Dimensions	4 m × 4 m	10 m × 4 m
Nominal maximum payload	60 t	100 t
Frequency range	0.01 ÷ 60 Hz	0.01 ÷ 60 Hz
Max displacement	X and Y: ±0.4 m, Z: ±0.25 m	X and Y: ±0.4 m, Z: ±0.25 m
Max velocity	X and Y: ±2.2 m/s, Z: ±1.5 m/s	X and Y: ±1.1 m/s, Z: ±0.75 m/s
Max acceleration (at max payload)	X and Y: ±1.5 g, Z: ±1.0 g	X and Y: ±1.05 g, Z: ±0.7 g

**Table 3 sensors-25-01010-t003:** Comparison of sensor properties.

Company	Product	Type	N. of Axes	Noise Floor [µg/√Hz]	Sensitivity [mV/g]	Sensitivity [µg/LSB]	Full-Scale Range Dynamic Range	Bandwidth
Seiko-Epson	M-A352 ^(1)^	Digital 32-bit MEMS	3	0.2 at 0.5–30 Hz		0.06	±15 g > 140 dB	0–460 Hz
Safran-Colibyis	SI1003	Analog MEMS	1	0.7	900		±3 g 108.5 dB (0.1–100 Hz)	0–500 Hz
Safran-Colibrys	VS1002 ^(1)^	Analog MEMS	1	7	1350		±2 g 108.5 dB (0.1–100 Hz)	0–700 Hz
Analog Device	ADXL355 ^(1)^	Digital 20-bit MEMS	3	22.5 Hz at ±2 g		3.9 at ± 2 g	±2 g to ±8 g~90 dB (±2 g)	1–1000 Hz
PCB Piezotronics	3711B1110G ^(2)^	Analog MEMS	1	107.9	1000		±10 g	0–1000 Hz
PCB Piezotronics	393B04 ^(1)^	Analog piezoelectric	1	0.3 at 1 Hz 0.1 at 10 Hz	1000		±5 g	0.06–450 Hz
PCB Piezotronics	T333B50 ^(1)^	Analog piezoelectric	1	15 at 1 Hz 3.8 at 10 Hz 1.1 at 100 Hz	1000		±5 g	0.5–3000 Hz
PCB Piezotronics	356A17 ^(1)^	Analog piezoelectric	3	18 at 1 Hz 6 at 10 Hz 2 at 100 Hz	500		±10 g	0.5–3000 Hz
Kinemetrics	Episensor ES-T	Force balance	3	0.06 at 1 Hz (±0.25 g)	10,000		±0.25 g to ±4 g 155 dB	DC–200 Hz

^(1)^ Installed on the specimen or on the shake table; ^(2)^ Installed on the shake table for system control.

**Table 4 sensors-25-01010-t004:** Position of the sensors on the structure and adopted sampling rates. In the table, the hyphen (-) is adopted if the position is not definable by F, L or V.

Graphical Representation	Station Code	F	L	V	Model	Sampling Rate [Hz]
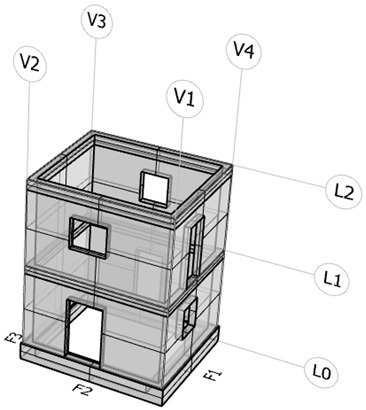 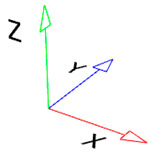	M1	3	1	3	M-A352	200
	M2	3	2	2	M-A352	200
	M3	1	1	4	M-A352	200
	M4	3	2	3	M-A352	200
	M5	1	2	1	M-A352	200
	M6	1	1	1	M-A352	200
	M7	3	1	2	M-A352	200
	M8	-	0	-	M-A352	200
	M9	1	2	4	M-A352	200
	M10	2	1	2	ADXL355	125
	M11	2	1	1	ADXL355	125
	M12	-	0	-	ADXL355	125
	W1	2	1	1	VS1002	1000
	W2	2	1	2	VS1002	1000
	W3	4	1	3	VS1002	1000
	W4	4	1	4	VS1002	1000
	W5	2	2	1	VS1002	1000
	W6	2	2	2	VS1002	1000
	W7	4	2	3	VS1002	1000
	W8	4	2	4	VS1002	1000
	A1	1	1	4	356A17	1000
	A2	1	1	1	356A17	1000
	A3	3	1	2	356A17	1000
	A4	3	1	3	356A17	1000
	A5	1	2	4	356A17	1000
	A6	1	2	1	356A17	1000
	A7	3	2	2	356A17	1000
	A8	3	2	3	356A17	1000
	F2	2	1	-	T333B50	1000
	F3	3	1	-	T333B50	1000
	F4	4	1	-	T333B50	1000
	F6	2	2	-	T333B50	1000
	F7	3	2	-	T333B50	1000
	F8	4	2	-	T333B50	1000
	S1	0	1	1	393B04	1000
	S2	0	2	1	393B04	1000
	AccGDL	-	Shake table	-	3711B1110G	1000

**Table 5 sensors-25-01010-t005:** Sequence of the actions performed on the shaking table during the test.

Type of Excitation	Intensity [g]	Direction
WN	0.03 (RMS)	X
WN	0.04 (RMS)	X
WN	0.03 (RMS)	Y
WN	0.04 (RMS)	Y
WN	0.015 (RMS)	Z
WN	0.03 (RMS)	Z
WN	0.04 (RMS)	XYZ
EQ	0.05 (PGA)	XYZ
WN	0.03 (RMS)	XYZ
EQ	0.1 (PGA)	XYZ
WN	0.03 (RMS)	XYZ
EQ	0.2 (PGA)	XYZ
WN	0.03 (RMS)	XYZ
EQ	0.3 (PGA)	XYZ
WN	0.03 (RMS)	XYZ
EQ	0.4 (PGA)	XYZ
WN	0.03 (RMS)	XYZ
EQ	0.5 (PGA)	XYZ
WN	0.03 (RMS)	XYZ

**Table 6 sensors-25-01010-t006:** Fundamental frequency estimation.

	Formula	C	a	Fundamental Frequency *f* [Hz] of the Full-Scale Building	Fundamental Frequency *f* [Hz] of the Specimen
Alguhane et al. [22] (D = 2.96 m)	$f=\frac{1}{C\cdot H\cdot D^{-\alpha }}$	0.07	0.5	4.91	6.01
Alguhane et al. [22] (D = 3.41 m)	$f=\frac{1}{C\cdot H\cdot D^{-\alpha }}$	0.07	0.5	5.27	6.46
EUROCODE 8 [18]-NTC18 [23]	$f=\frac{1}{C\cdot H^{\alpha }}$	0.05	0.75	5.13	6.29
ASCE 7-10 [24]	$f=\frac{1}{C\cdot H^{\alpha }}$	0.0488	0.75	5.26	6.44
Alguhane et al. [22]	$f=\frac{1}{C\cdot H^{\alpha }}$	0.042	0.75	6.11	7.49

**Table 7 sensors-25-01010-t007:** Modal shape and frequency comparison.

Sensor	Mode 1	Mode 2	Mode 3
M-A352 before the hydraulic pumps’ activation	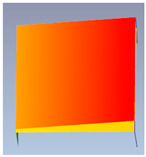 7.584 Hz	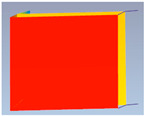 8.301 Hz	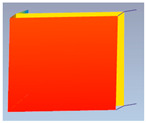 9.823 Hz
M-A352 during the hydraulic pumps’ activation	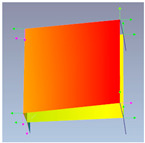 7.050 Hz	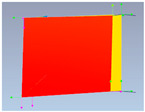 7.549 Hz	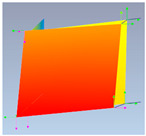 9.875 Hz
PCB356A17during the hydraulic pumps’ activation	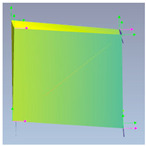 6.871 Hz	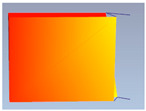 7.208 Hz	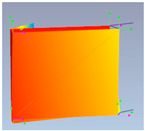 8.768 Hz

**Table 8 sensors-25-01010-t008:** MAC between modes calculated before and during the hydraulic pumps’ activation.

		M-A352 Before the Hydraulic Pumps’ Activation		
		7.584 Hz	8.301 Hz	9.823 Hz
M-A352 during the hydraulic pumps’ activation	7.050 Hz	0.9411		
	7.549 Hz		0.829	
	9.875 Hz			0.8339

**Table 9 sensors-25-01010-t009:** MAC between modes calculated during the hydraulic pumps’ activation.

		M-A352 During the Hydraulic Pumps’ Activation		
		7.050 Hz	7.549 Hz	9.875 Hz
PCB356A17 during the hydraulic pumps’ activation	6.871 Hz	0.4955		
	7.208 Hz		0.419	
	8.768 Hz			0.344

**Table 10 sensors-25-01010-t010:** First frequency variation.

Chunk of Data	Frequency f1 [Hz]	Damping [%]	Complexity [%]
01_R1345_1355	6.98	4.44	4.34
Seismic action at 10% of the PGA max			
04_R1407_1417	6.46	8.05	17.47
Seismic action at 20% of the PGA max			
07_R1433_1444	6.44	6.30	33.61
Seismic action at 30% of the PGA max			
09_R1445_1451	5.75	6.72	26.66
Seismic action at 40% of the PGA max			
12_R1500_1505	5.001	0.207	23.868
Seismic action at 50% of the PGA max			
15_R1514_1520	5.014	7.468	40.055

**Table 11 sensors-25-01010-t011:** Compared sensors. In the table, the hyphen (-) is adopted if the sensor does not acquire data in a certain direction.

Station Code	F	L	V	Noise Density [µg/√Hz]	Sampling Rate [Hz]	X	Y	Z
M6	1	1	1	0.2	200	z	x	y
M5	1	2	1	0.2	200	z	x	y
W1	2	1	1	7	1000	x	y	-
W5	2	2	1	7	1000	x	y	-
M11	2	1	1	25	200	x	-z	y
A2	1	1	1	18-6-2 @ 1-10-100 Hz	1000	X	Y	Z
A6	1	2	1	18-6-2 @ 1-10-100 Hz	1000	X	Y	Z

## Data Availability

The data presented in this study are available on request from the corresponding author.

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
