# Peer review of "Shake Table Tests on Scaled Masonry Building: Comparison of Performance of Various Micro-Electromechanical System Accelerometers (MEMS) for Structural Health Monitoring"

_sensors, 2025, doi:10.3390/s25041010_

Round 1

Reviewer 1 Report

Comments and Suggestions for Authors

Major comments: 

1. The study presents a comprehensive investigation into the performance of various MEMS accelerometers in a scaled masonry building under seismic testing. However, the methodology could be enhanced by providing more detailed descriptions of the testing parameters, such as the specific configurations of the sensors and the exact conditions under which they were tested (e.g., ambient noise levels). 

2. The authors should emphasize how their findings contribute to existing literature on MEMS accelerometers compared to traditional sensors, particularly in low-excitability structures like masonry buildings. A clearer statement on what differentiates this study from previous research would strengthen its impact. 

3. Moreover, the authors fail to discuss the recent advancements in sensor-based SHM for real-time applications. Online SHM approaches such as eigen perturbation techniques have demonstrated excellent adequacy in detecting real-time damage, identifying system parameters, and providing information about remaining useful life. The authors should clearly mention the justification of the proposed approach and compare the computational efficacy against recently established eigen perturbation approaches. 

4. Has the proposed approach shown to perform using just a single sensor? The authors should justify the inclusion of this technique in the literature when real-time approaches - such as the ones involved in eigen perturbation - have shown to perform such using single sensors. 

5. The motivation behind using a scaled model and comparing different sensor types is well-founded; however, it would benefit from a more robust discussion on the implications of these findings for real-world applications.

Minor comments: 

1. In Section 1, "which act as an unknown input" should be revised to "which acts as an unknown input" for subject-verb agreement. 

2. In Section 2.1, "the structure is bonded with an M5 category pre-mixed mortar" should be corrected to "the structure is bonded with M5 category pre-mixed mortar" to avoid unnecessary articles. 

3. The following figures lack clarity in terms of DPI resolution and seem to be screenshots obtained from other sources: Figs. 2, 5, 7, 8, 9-16. These should be modified in this round of revisions. 

Author Response

Reviewer #1:

The authors wish to thank the reviewer for his/her comments. In the following a detailed point-by-point response is reported. All the amendments are green highlighted in the paper.

Comment #1 - The study presents a comprehensive investigation into the performance of various MEMS accelerometers in a scaled masonry building under seismic testing. However, the methodology could be enhanced by providing more detailed descriptions of the testing parameters, such as the specific configurations of the sensors and the exact conditions under which they were tested (e.g., ambient noise levels). 

RESPONSE: Thanks to the reviewer for the comment. The following sentence has been added to the paragraph 3. Results:

“We preliminary performed a comparative analysis of the average noise spectra recorded one month before the test (October 19, 2024), during a period of relative quiet, with the shake table motors turned off, and during the test itself (November 22, 2024) with the shake table motors in operation. The results showed that vibrations induced by the shake table were localized in the lower frequency range (1.8–2.6 Hz) and did not significantly affect the overall environmental noise conditions for the OMA analysis.”

Comment #2 - The authors should emphasize how their findings contribute to existing literature on MEMS accelerometers compared to traditional sensors, particularly in low-excitability structures like masonry buildings. A clearer statement on what differentiates this study from previous research would strengthen its impact. 

RESPONSE: The authors thank the reviewer for highlighting this important aspect of the study. We have added some statements in the conclusion to address how our findings contribute to existing literature. Specifically, we emphasize that the aim of our study was not to advocate for the use of MEMS accelerometers over traditional sensors but to compare the performance and accuracy of different sensor types, particularly in low-excitability structures like masonry buildings. While both MEMS accelerometers and traditional sensors are commonly used in structural health monitoring (SHM) by professionals and private companies, our focus was to assess their reliability and provide insights that would help to make informed decisions when selecting sensors for SHM and campaign. 

Comment #3 - Moreover, the authors fail to discuss the recent advancements in sensor-based SHM for real-time applications. Online SHM approaches such as eigen perturbation techniques have demonstrated excellent adequacy in detecting real-time damage, identifying system parameters, and providing information about remaining useful life. The authors should clearly mention the justification of the proposed approach and compare the computational efficacy against recently established eigen perturbation approaches. 

RESPONSE: The authors appreciate the reviewer's deep knowledge in this field. However, the paper was not specifically oriented towards damage detection strategies. No specific damage detection method was proposed in the paper; instead, the authors focused on the reliability of accelerometric measures. A more focused work on damage detection and real-time structural health monitoring (SHM) is one of the aims for future developments.

Comment #4 - Has the proposed approach shown to perform using just a single sensor? The authors should justify the inclusion of this technique in the literature when real-time approaches - such as the ones involved in eigen perturbation - have shown to perform such using single sensors. 

RESPONSE: The authors would like to clarify that the analyses were performed using four sensor typologies, as shown in Figure 9. The aim of the research was not to investigate damage detection algorithms. The eigen perturbation is outside the scope of the paper. The authors hope this clarifies any misunderstandings, but they are open to introducing additional statements if the reviewer considers that they may enhance the quality of the paper.

Comment #5 - The motivation behind using a scaled model and comparing different sensor types is well-founded; however, it would benefit from a more robust discussion on the implications of these findings for real-world applications.

RESPONSE: The conclusions have been enhanced on the base of the reviewer’s comment underling the impact of the research on the real word. The following sentences have been added to the conclusions, starting from line 611:

“The main aim of our work was to evaluate the advantages and disadvantages of these sensors, focusing on some critical characteristics such as sensitivity and self-noise. Then the performance of a variety of accelerometer sensors—namely Seiko-Epson M-A352, Safran-Colybris VS1002, Analog Devices ADXL355, PCB Piezotronics T333B50, 393B04, and 356A17—has been compared in their application to a scaled real structure subjected to both ambient and seismic excitations. Due to the reduced value of the scale factor (2:3) the results can be considered valid for both the actual and scaled building since their structural main frequencies, although higher in the scaled specimen, are still in the working range of the used sensors.

It is important to note, however, that evaluating the accuracy of sensors in capturing the system’s response under normal operating conditions and seismic events is just one critical aspect to consider in SHM. The success of SHM systems in real structures relies not only on the careful selection of sensors but also on their proper installation, effective data transmission methods, and efficient data processing strategies, beyond device and sensor maintenance, data volume management, and processing time.”

Minor comments: 

Comment #6 - In Section 1, "which act as an unknown input" should be revised to "which acts as an unknown input" for subject-verb agreement. 

RESPONSE: The text has been amended.

Comment #7 - In Section 2.1, "the structure is bonded with an M5 category pre-mixed mortar" should be corrected to "the structure is bonded with M5 category pre-mixed mortar" to avoid unnecessary articles. 

RESPONSE: The text has been amended.

Comment #8 - The following figures lack clarity in terms of DPI resolution and seem to be screenshots obtained from other sources: Figs. 2, 5, 7, 8, 9-16. These should be modified in this round of revisions.

RESPONSE: The authors confirm the accuracy of the figures in the word version. Possible DPI reduction may affect the PDF version.

Reviewer 2 Report

Comments and Suggestions for Authors

The paper presents an experimental study to evaluate the performance of different types of accelerometers to measure/monitor the modal properties and seismic response of a masonry specimen. The following clarifications and improvements are recommended, before accepting the paper for publishing, aiming at improving its quality and interest for the scientific community:

1.     Page 3 Line 104: “…2:3 using Froude similarity…”. Was only Froude similitude law adopted? Or both Cauchy and Froude's similitude laws?

2.     Figure 1 and Figure 2: Add the units of the dimensions.

3.     Page 5: Additional masses associated with the structural elements were added to respect the similitude law (3553 kg). Then, masses to simulate 50% of the non-structural loads (G2) and variable loads (Q) were also added. Were these loads (G2 and Q) scaled to take also into account the similitude law? For the seismic action, the load combinations defined in the Eurocode 8 should be adopted. Why was 50% of the loads adopted for both types of load?

4.     Table 3: The VS1002 accelerometer has very high sensitivity (10 v/g) and noise (7 μg/ √ Hz). Are these features corrected? Three types of piezoelectric accelerometers with two different sensitivities were used (1 V/g and 0.5 V/g). It is noted that there are piezoelectric accelerometers from PCB with very high sensitivity (10 V/g), which are more appropriate for dynamic identification tests with low amplitude action/ambient vibration. Where was the Episensor ES-T used?

5.     Table 4: The accelerometers A1 and A2 are exactly in the same position. All the A accelerometers have the same V position. Review the position definition of all accelerometers in the table. Why the piezoelectric accelerometer with the lowest sensitivity (0.5 V/g) was adopted to measure the vibrations of the specimens? Why the same number of the different accelerometers, located at the same positions on the specimen, was not adopted? This test setup would allow to do a direct and totally fair comparison of the performance of the different types of accelerometers for the response at each point of the specimen and for the global parameters (frequencies and mode shapes).

6.     Table 5: Does the intensity correspond to the PGA or RMS?

7.     Page 12 Section 3.1: Why was the HVSR analysis not done for all the accelerometer types, allowing the comparison of the performance of the different accelerometer types?

8.     Page 13 Section 3.2: It is recommended to improve the discussion of the different accelerometer types presenting and discussing in more detail the results obtained from the dynamic identification tests (all the modes of the specimen, including the respective frequencies variations and differences in the mode shapes using the MAC values).

9.     Page 13 Line 349: Replace the number of the reference 45 by 22.

10. Page 14 Line 353: “…structure under examination is not a full-scale building…”. The first frequency can be firstly estimated at full scale and then scaled based on the similitude law.

11. Page 14 Line 382: The first frequency estimated from OMA is 10% higher than the one estimated under hydraulic pump noise.  Does the specimen present initial damage/cracks, which can be opened for a higher excitation and consequently reduce the frequency?

12. Page 18 Section 3.3: It is recommended to improve this Section focusing more on the direct comparison of the performance of the different accelerometers, which is the main aim of the paper, and less on the response variation of the specimen for the seismic action.

13. Page 25 Conclusions: It is known that accelerometers with high sensitivity (for example, 10 V/g) are the most appropriate to do dynamic identification tests with low amplitude, such as ambient vibration tests (very low accelerations). On the other hand, accelerometers with a high amplitude range and consequently lower sensitivity are more appropriate for seismic tests (higher accelerations). The tested specimen was built at reduced scale, which has influence on the frequencies of the modes, and consequently on the conclusions of the performance of the different types of accelerometers for the full scale structure. At full scale, and considering the Chauch and Froude similitude laws, the structure presents lower frequencies, which are easier to estimate experimentally. Thus, some accelerometers can present a poor performance in estimating the frequencies at reduced scale (high frequencies) and a better performance in estimating the frequencies at reduced scale (lower frequencies). This aspect should be discussed and clarified in the Conclusions.

Author Response

Reviewer #2:

The paper presents an experimental study to evaluate the performance of different types of accelerometers to measure/monitor the modal properties and seismic response of a masonry specimen. The following clarifications and improvements are recommended, before accepting the paper for publishing, aiming at improving its quality and interest for the scientific community:

The authors wish to thank the reviewer for having appreciated the paper and for all the comments and suggestions aimed at improving the final version of the manuscript. In the following a detailed response to each suggested point is reported. All the amendments are yellow highlighted in the paper.

Comment #1 - Page 3 Line 104: “…2:3 using Froude similarity…”. Was only Froude similitude law adopted? Or both Cauchy and Froude's similitude laws?

RESPONSE: The experimental model has been built by considering only Froude similitude law as suggested in the cited literature [12]. Indeed, this is customary for shaking table testing when both the used material properties (elastic modulus) and acceleration (since gravity acceleration affects the dynamic behaviour of both the prototype and the model) do not change between the real scale prototype and the scaled model.

Comment #2 - Figure 1 and Figure 2: Add the units of the dimensions.

RESPONSE: The units have been specified in the captions.

Comment #3 - Page 5: Additional masses associated with the structural elements were added to respect the similitude law (3553 kg). Then, masses to simulate 50% of the non-structural loads (G2) and variable loads (Q) were also added. Were these loads (G2 and Q) scaled to take also into account the similitude law? For the seismic action, the load combinations defined in the Eurocode 8 should be adopted. Why was 50% of the loads adopted for both types of load?

RESPONSE: The selected loads are not intended to verify the seismic performance of the structure in under investigation but to study its dynamic behaviour under increasing seismic inputs and to compare the performance of the different types of sensors used. Nevertheless, both the masses and the loads G2 and Q were scaled considering Froude's similarity law, and their values were considered in line with those typically used in historical buildings in the considered area and according to the Italian Standard NTC2018. Table 1 and the text on Section 2.1 have been modified and the following sentence has been introduced: “a load equal to (G2 + Y2Q)×A was considered in order to take into account actual values of loads according to local historical buildings and the Italian seismic code.”

Comment #4 - Table 3: The VS1002 accelerometer has very high sensitivity (10 v/g) and noise (7 μg/ √ Hz). Are these features corrected? Three types of piezoelectric accelerometers with two different sensitivities were used (1 V/g and 0.5 V/g). It is noted that there are piezoelectric accelerometers from PCB with very high sensitivity (10 V/g), which are more appropriate for dynamic identification tests with low amplitude action/ambient vibration. Where was the Episensor ES-T used?

RESPONSE

We have corrected the error in the table regarding the sensitivity of the VS1002, which is not 10 V/g but rather 1359 mV/g as reported by manufacture. The EpiSensor ES-T, one of the most commonly used accelerometers in seismology, was not included in the experiment. However, it has been listed in the table alongside the Safran Colybris SI1003 and reported in figure 5 where self-noises are shown, solely for the purpose of comparing their characteristics with the other sensors used in the study. Only the sensors with single quote mark (1) were placed on the specimen. We acknowledge that there are piezoelectric accelerometers from PCB, such as the 393B31 and 393B12, with very high sensitivity (10 V/g), which are particularly suitable for dynamic identification tests involving low-amplitude actions or ambient vibrations. Unfortunately, an adequate number of these sensors was not available for the study.

Comment #5 - Table 4: The accelerometers A1 and A2 are exactly in the same position. All the A accelerometers have the same V position. Review the position definition of all accelerometers in the table. Why the piezoelectric accelerometer with the lowest sensitivity (0.5 V/g) was adopted to measure the vibrations of the specimens? Why the same number of the different accelerometers, located at the same positions on the specimen, was not adopted? This test setup would allow to do a direct and totally fair comparison of the performance of the different types of accelerometers for the response at each point of the specimen and for the global parameters (frequencies and mode shapes).

RESPONSE: The positions of all the A accelerometers have been corrected in Table 4. The vibrations of the specimen have been acquired with several accelerometers for comparison of performances under different excitation intensities. Most of them have been placed to acquire the acceleration response at approximately same positions: in the opinion of the authors each corner of the reinforced concrete floor curbs can be considered stiff enough and the same acceleration can be recorded, for example, at position F1-L1-V1 (sensors A2 and M6) and F2-L1-V1 (Sensors M11 and W1) and then for each cluster of sensors the results can be considered comparable.

With respect to the question on the piezoelectric accelerometers named with Ax, they were the only triaxial ICP commercial accelerometers available at the laboratory and it has been showed that they perform very well under random noise and seismic excitations. For such a reason they have been used to compute the natural frequencies and mode shapes of the structure that have been used as a reference for comparison with the high sensitivity MEMS accelerometers. On the other hand, Ax accelerometers did not perform adequately in the case of Ambient Vibration Tests (AVT) at very low input intensity when the high sensitivity MEMS accelerometers showed a good performance, and the modal properties computed are in good agreement with the one computed by the commercial Ax accelerometers.

Comment #6 -   Table 5: Does the intensity correspond to the PGA or RMS?

RESPONSE: The table 5 has been updated and the intensities have been specified.

Comment #7 -    Page 12 Section 3.1: Why was the HVSR analysis not done for all the accelerometer types, allowing the comparison of the performance of the different accelerometer types?

RESPONSE: Thank you for your comment. The primary objective of the HVSR analysis conducted during the experiments, as described in our study, was to characterize the behaviour (response and amplification) of the structure   on the shaking table rather than to evaluate or compare the performance of different measuring instruments.

Although not included in the manuscript, a comparison of the H/V curves obtained from the accelerometric sensors Seiko-Epson M-A352 and the more sensitive and lower self-noise three-component velocimetric sensor ETL3D/5s (Fertitta et al., 2020), both co-located and installed at level 0, showed consistent results. Consequently, we determined that calculating the H/V ratio with all sensor types would not provide additional insights into the structure's response or amplification characteristics. (Fertitta, G.; Costanza, A.; D’anna, G.; Patanè, D. The Earth Lab 5s (ETL3D/5s) seismic sensor. Design and test. Ann. Geophys. 2020, 63, 2.)

Comment #8 - Page 13 Section 3.2: It is recommended to improve the discussion of the different accelerometer types presenting and discussing in more detail the results obtained from the dynamic identification tests (all the modes of the specimen, including the respective frequencies variations and differences in the mode shapes using the MAC values).

RESPONSE: According to the reviewer’s suggestion the section 3.2 has been improved with additional comparison and MAC values, starting from line 413.

Comment #9 -   Page 13 Line 349: Replace the number of the reference 45 by 22.

RESPONSE: The reference has been updated.

Comment #10 - Page 14 Line 353: “…structure under examination is not a full-scale building…”. The first frequency can be firstly estimated at full scale and then scaled based on the similitude law.

RESPONSE: The authors thank the reviewer for the comment. Table 6 has been modified according to the suggestion.

Comment 11 - Page 14 Line 382: The first frequency estimated from OMA is 10% higher than the one estimated under hydraulic pump noise.  Does the specimen present initial damage/cracks, which can be opened for a higher excitation and consequently reduce the frequency?

RESPONSE: As figure 16 reports, limited cracks were present before starting the shaking table tests. The authors supposed that the difference was caused by different temperature during the two acquisitions, as already mentioned in the paper.

Comment #12 - Page 18 Section 3.3: It is recommended to improve this Section focusing more on the direct comparison of the performance of the different accelerometers, which is the main aim of the paper, and less on the response variation of the specimen for the seismic action.

RESPONSE: The authors thank the reviewer for the comments and improved the section 3.2 with consideration that account for a better comprehension of the comparisons. A sentence has been introduced at the end of the section 3.3 line 560.

Comment #13 -   Page 25 Conclusions: It is known that accelerometers with high sensitivity (for example, 10 V/g) are the most appropriate to do dynamic identification tests with low amplitude, such as ambient vibration tests (very low accelerations). On the other hand, accelerometers with a high amplitude range and consequently lower sensitivity are more appropriate for seismic tests (higher accelerations). The tested specimen was built at reduced scale, which has influence on the frequencies of the modes, and consequently on the conclusions of the performance of the different types of accelerometers for the full scale structure. At full scale, and considering the Chauch and Froude similitude laws, the structure presents lower frequencies, which are easier to estimate experimentally. Thus, some accelerometers can present a poor performance in estimating the frequencies at reduced scale (high frequencies) and a better performance in estimating the frequencies at reduced scale (lower frequencies). This aspect should be discussed and clarified in the Conclusions.

RESPONSE: Thanks to the reviewer for the comment and suggestions. Indeed, the authors agree with the comments of the reviewer, but it is worth to remark that in the specific case study, since the scale factor is quite close to unity, the variation in the range of characteristic frequencies between the full-scale model and the reduced scale one is not such as to justify a different behaviour of the sensors used. In fact, all the accelerometers deployed have operating ranges that cover the main vibration frequencies of both the hypothetical full-scale structure and the tested scale model. For this reason, the authors chose to use this case study also for comparing the performance of various sensors used.

Round 2

Reviewer 1 Report

Comments and Suggestions for Authors

The authors have adequately addressed mt review queries. I accept the paper at this stage. 

Reviewer 2 Report

Comments and Suggestions for Authors

The Authors have clarified all the aspects referred in the comments and the paper was improved. Thus, the manuscript is recommended for publishing.